

# Exposure to mass media family planning messages among men in Nigeria: analysis of the Demographic and Health Survey data

Daniel Amoak[1], Irenius Konkor[2], Kamaldeen Mohammed[1], Sulemana Ansumah Saaka[1] and Roger Antabe[3]

[1] University of Western Ontario, London, Ontario, Canada
[2] University of Toronto, Mississauga, Ontario, Canada
[3] University of Toronto, Scarborough, Ontario, Canada

Corresponding author
Daniel Amoak, damoak@uwo.ca

## ABSTRACT

**Background**. Family planning (FP) is essential for improving health and achieving reproductive goals. Although men are important participants in FP decision-making within households in Nigeria, a country with one of the highest rates of maternal mortality, we know very little about their exposure to mass media FP messages.

**Methods**. Drawing theoretical insights from the structural influence model of health communication and using the 2018 Nigeria Demographic and Health Survey ($n = 13,294$), and applying logistic regression analysis, we explored the factors associated with men's exposure to mass media FP messages in Nigeria.

**Results**. A range of socioeconomic, locational, and demographic factors were associated with men's exposure to mass media FP messages. For example, wealthier, more educated, and employed men were more likely to be exposed to mass media FP messages than their poorer, less educated, and unemployed counterparts. In addition, compared to those in rural areas and other regions, men in urban areas as well as South East Region, were more likely to be exposed to mass media FP messages. Finally, younger men and those who belong to the traditional religion were less likely to be exposed to mass media FP messages, compared to their older and Christian counterparts.

**Conclusions**. Based on these findings, we discuss implications and recommendations for policymakers as well as directions for future research.

## INTRODUCTION

Family planning (FP) is essential for improving health and achieving reproductive goals (*Kantorová et al., 2020*). Several studies have found that FP can help reduce unwanted pregnancies, pregnancy-related complications (*e.g.*, hemorrhage, sepsis, obstructed labour, and reproductive cancers), and maternal deaths (*Bongaarts, 2011*; *Khan et al., 2006*; *World Health Organization, 2022*). Despite the critical role of FP, nearly 214 million women in developing countries do not have access to FP, with the greatest unmet need seen in

sub-Saharan Africa (SSA) (*Tessema et al., 2016*). In SSA, approximately 17% of women have unmet needs for FP, which is considerably higher than the global average of 10% (*United Nations Department of Economic and Social Affairs, 2019*).

In the last two decades, many developing countries including those in SSA have made substantive progress in reducing unwanted pregnancies and pregnancy-related complications, resulting in decreased maternal deaths. This progress, however, has been uneven within the sub-region. For instance, Nigeria's maternal mortality rate is among the highest worldwide. Recent evidence suggests that Nigeria has the fourth worst maternal mortality ratio in the world with 917 for every 100,000 live births (*Onoja et al., 2022*). More concerning is the fact that 44% of deaths were reported among marginalized and structurally exposed groups in the country (*Badamasi, 2021*). In this context, the persistent low uptake of FP may be a critical determinant of adverse health outcomes including high maternal mortality rates, making it important to identify ways that may be useful to promote and enhance the uptake of FP in Nigeria (*Federal Government of Nigeria, 2020*).

Mass media communication has been identified as a valuable tool in the promotion of healthy sexual and reproductive behaviours including the uptake of FP (*Cahill et al., 2018*; *Dougherty et al., 2018*; *World Health Organization, 2022*). For example, the Federal Ministry of Health, through the Nigeria Family Planning Blueprint 2020–2024, has outlined the use of mass media such as television, radio, newspaper, magazine and text messaging to reach a wide range of audience on the need for the uptake of FP (*Federal Government of Nigeria, 2020*). To augment the efforts of the federal government, other non-profit agencies such as the Nigerian Urban Reproductive Health Initiative are also intensifying FP messaging through mass communication. In the context of SSA, including Nigeria, the key role of mass media communication in health messaging is hinged on a number of structural factors. Specifically, mass media communication has the ability to decode complex FP messages generated at the policy level into simple messages often translated into the local dialect for easy digestion by the local populates (*Kansanga et al., 2018*). This is particularly important in Nigeria where the health care system which also serves as a source of health information may not be fully accessible, especially for rural residents and other structurally disadvantaged people (*Abubakar et al., 2022*).

Supporting this argument, studies have identified the importance of women's mass media exposure on their uptake of FP in SSA. For example, in Ghana, it was found that women who reported being exposed to FP messages (adverts, brands) on television and radio were more likely to practice FP methods including barrier methods, and hormonal methods than those who were not (*Appiah et al., 2020*). In the Nigerian context, *Chima & Alawode (2019)* highlighted that increased exposure to mass media messaging, including radio and television was positively associated with the use of FP. Among female adolescents in rural Nigeria, it was also found that exposure to FP messaging on television and radio was associated with higher likelihood of modern contraceptive use (*Chima & Alawode, 2019*).

Although these studies are useful, they are exclusively focused on women, largely overlooking men's roles in reproduction decision-making in the context of FP mass media messaging. For instance, studies have shown that the effectiveness of women's

uptake of FP partly depends on the household decision-making dynamics, which makes men's knowledge of FP and exposure to FP mass media messaging critical (*Aboagye et al., 2021*; *Antabe, Sano & Luginaah, 2022*; *Sano et al., 2018*). Indeed, limited evidence in SSA including Nigeria suggests that when men were exposed to FP mass media messaging with their partners, the uptake of FP increased among the couple but uptake remained the same when only women were exposed to FP mass media messaging (*Ijadunola et al., 2010*). Similarly, *Mutumba (2022)* found that exposure to FP mass media messaging increased FP knowledge, attitudes, and adoption among men in SSA. These findings demonstrate that the involvement of men is crucial for the success of any FP mass media intervention. This is more compelling in a patriarchal context like Nigeria where gender, sociocultural norms and household decision-making power dynamics may often favour men.

Using a nationally representative survey and applying the structural influence model of health communication, we aim to address this void in the literature by identifying potential mechanisms by which intermediary and structural factors are associated with men's exposure to mass media FP messages in Nigeria. Findings from this study may be helpful for informing stakeholders to take necessary steps to implement programs to increase the uptake of FP as part of important national-level policy initiatives such as the Nigeria Family Planning Blueprint 2020–2024 and Nigeria FP2030 Commitment.

## Structural influence model of health communication: the context of Nigeria

This article draws theoretical insights from the structural influence model of health communication, which posits that inequalities in access to health communication such as mass media FP messages are generated through the interplay of structural and mediating/moderating factors (*Viswanath, Wallington & Blake, 2009*). The theory contends that structural factors such as socioeconomic and geographical characteristics exert overarching influence on people's access to health information through mass media in a given context.

For example, educational attainment is a critical determinant of exposure to mass media health communication in at least two ways. For one, people with high educational attainment may be more conscious about their health and are known to voluntarily seek health information to make informed health-related decisions for themselves and respective households (*Do et al., 2020*). For another, in Nigeria where there is a high level of illiteracy, educated people may be better positioned to digest health information presented through mass media, as such messages may only be delivered in the official English language, which makes it inaccessible to people without formal education (*Azees et al., 2022*).

The theory also identifies household wealth and employment to create an opportunity for individuals and households to access large spectrum of mass media communication outlets as it removes any associated financial burdens with accessing these mass media platforms. This may particularly be the case in Nigeria where the income-to-cost ratio in purchasing basic mass media devices such as television, radio, and subscription to newspapers and magazines may offset household budgets as the priority may be on meeting basic household needs such as shelter and food (*Amzat, 2011*).

In addition to these socioeconomic factors, locational characteristics including region of residence and rural–urban residency may influence access to mass media communication in Nigeria (*Ajaero et al., 2016*). Specifically, the concentration of mass media infrastructure in Southern Nigeria may work to increase exposure of residents in this region relative to those in the north (*Chima & Alawode, 2019*). Similarly, due to urban bias in the location of social infrastructure availability including mass media, urban residents may tend to have advantage in accessing and being exposed to health communication through a wider range of mass media outlets (*Abubakar et al., 2022*).

Beyond structural factors, the theory points to the role of mediating/moderating factors. These factors often include demographic characteristics such as age, marital status, and religious affiliation. For example, research shows that people's increased need for health care due to aging may work to heighten their health-seeking behaviours including accessing health communication through mass media (*Aboagye et al., 2021*). Moreover, it has been shown that there are unique health benefits linked to marriage where married couples do not only report better health outcomes but also often adapt preventative health behaviours that may include accessing mass media health communication (*Mutumba, 2022*). Finally, religious affiliation is a critical cultural factor in Nigeria that may influence health behaviours including seeking information on specific health conditions. For example, given traditional religion and Islam's perceived emphasis on high fecundity, it is possible that members of these groups may be discouraged from actively seeking information on the use of FP *via* various sources including mass media (*Barro & Bado, 2021*). These mediating/moderating factors may interplay with broader structural factors to shape men's exposure to mass media FP messages in Nigeria.

## MATERIALS & METHODS

### Data

We used the data from the 2018 Nigeria Demographic and Health Survey (NDHS), which is a nationally representative survey of Nigerian men aged 15–59 and women aged 15–49. The NDHS was implemented by the National Population Commission in collaboration with the National Malaria Elimination Programme of the Federal Ministry of Health, with technical assistance from ICF through the DHS Program. The NDHS provides high quality and reliable information on basic demographic indices and health-related topics including exposure to mass media FP messages. The NDHS used a multi-stage sampling framework where systematic sampling with probability proportional to size was applied to identify enumeration areas from which households were chosen. While the NDHS also interviews women, this study exclusively focuses on men. The NDHS initially identified 13,422 men aged 15–59 and successfully interviewed 13,311 men, with a response rate of 99%. We employed listwise deletion technique to address missing cases, as they accounted for less than 1% of the sample. To this end, our analytical sample includes 13,294 men who answered questions on exposure to mass media FP messages.

## Dependent variable

The NDHS asked respondents whether they have been exposed to FP messages on (1) radio, (2) television, (3) text messages, and (4) print media in the last few months. Based on these questions, we constructed a binary dependent variable called 'exposure to mass media FP messages' where respondents were coded 'yes' if they were exposed to FP messages on at least one of four media outlets (0 = no; 1 = yes). We decided to rely on this approach for two different reasons. For one, these four variables were highly correlated with robust internal consistency ($\alpha = 0.72$). Importantly, these four items were loaded into a single construct. For another, this approach is consistent with the policy implementation at the national level in Nigeria. Specifically, mass media messages on public health issues are structured such that people can obtain the same information through a range of mass media outlets such as television, radio, print media, and text messages (*Federal Government of Nigeria, 2020*).

## Independent variables

Informed by the structural influence model of health communication, we included two sets of independent variables. This model posits that there are two different sets of factors that may be influential to exposure to mass media messages, namely mediating/moderating and structural factors. For this study, we considered three mediating/moderating factors, such as age (0 = 55–59; 1 = 50–54; 2 = 45–49; 3 = 40–44; 4 = 35–39; 5 = 30–34; 6 = 25–29; 7 = 20–24; 8 = 15–19), marital status (0 = never married; 1 = currently married; 2 = formerly married), and religion (0 = Christian; 1 = Muslim; 2 = traditionalist). For structural factors, we include region of residence (0 = South East; 1 = South South; 2 = South West; 3 = North Central; 4 = North East; 5 = North West), place of residence (0 = urban; 1 = rural), education (0 = higher education; 1 = secondary education; 2 = primary education; 3 = no education), household wealth (0 = highest; 1 = higher; 2 = middle; 3 = lower; 4 = lowest), and employment status (0 = employed; 1 = unemployed).

## Statistical analysis

We employed three different analyses. First, we used univariate analysis to describe the characteristics of our analytical sample. Second, bivariate regression analysis was used to understand the gross impacts of the independent variables on the dependent variable. Finally, we conducted multivariate regression analysis to estimate the net impacts while simultaneously accounting for a range of mediating/moderating and structural factors. For regression analysis, we used logistic regression analysis due to the dichotomous nature of the dependent variable. Results were reported with odds ratio. Odds ratios (OR) larger than 1 indicate that men were more likely to have been exposed to mass media FP messages while those smaller than 1 point to lower odds of having been exposed. All analyses were carried out using STATA 17 (State Corp, College Station, TX, USA). The 'svy' function was applied in statistical analysis to adjust for the cluster sampling design as well as sampling weights.

## RESULTS

Table 1 shows descriptive characteristics of the study sample. We found that 55% of men had not been exposed to FP messages through at least one of the four mass media outlets (*i.e.,* radio, television, text messages, and print media) in the last few months. In terms of demographic characteristics, it was found that men aged 15–19 (18%) was the largest age group followed by 35–39 (14%) and 30–34 (13%). More than half of men were also currently married (53%) and lived in northern Nigeria (57%) and rural areas (54%). The largest religious group was Islam (62%) followed by Christian (37%) and traditionalist (1%). For socioeconomic characteristics, we found that about one quarter (23%) of men did not have any formal education although the majority (87%) were employed.

Table 2 shows findings from the bivariate analysis. Overall, we found that both mediating/moderating and structural factors were significantly associated with exposure to mass media FP messages. For example, it was found that men aged 15–19 (OR = 0.28, $p < 0.001$), 20–24 (OR = 0.52, $p < 0.001$), and 25–29 (OR = 0.74, $p < 0.05$) were less likely to have been exposed to mass media FP messages than those aged 55–59. We also found that currently married men were more likely to have been exposed to mass media FP messages than those who were never married (OR = 1.91, $p < 0.001$). For religion, compared to Christian men, Muslim (OR = 0.62, $p < 0.001$) and traditionalist (OR = 0.39, $p < 0.001$) men were less likely to have been exposed to mass media FP messages. Men in South South (OR = 0.45, $p < 0.001$), North Central (OR = 0.22, $p < 0.001$), North East (OR = 0.31, $p < 0.001$), and the North West (OR = 0.39, $p < 0.001$) regions were all less likely to have been exposed to mass media FP messages, compared to those in South East. Similarly, rural men were less likely to have been exposed to mass media FP messages than their urban counterparts (OR = 0.38, $p < 0.001$). Furthermore, men with secondary (OR = 0.42, $p < 0.001$), primary (OR = 0.39, $p < 0.001$), and no education (OR = 0.17, $p < 0.001$) were less likely to have been exposed to mass media FP messages than their counterparts with higher education. Similarly, men whose household income belongs the higher (OR = 0.67, $p < 0.001$), middle (OR = 0.40, $p < 0.001$), lower (OR = 0.28, $p < 0.001$), and lowest (OR = 0.13, $p < 0.001$) categories were less likely to have been exposed to mass media FP messages than those who belonged to the highest category. Unemployed men (OR = 0.46, $p < 0.001$) were also less likely to have been exposed to mass media FP messages compared with the employed men.

Multivariate findings are also shown in Table 2. We found that men aged 15–19 (OR = 0.29, $p < 0.001$), 20–24 (OR = 0.47, $p < 0.001$), 25–29 (OR = 0.64, $p < 0.001$), and 30–34 (OR = 0.72, $p < 0.01$) were less likely to have been exposed to mass media FP messages than those aged 55–59. Interestingly, the significant impact of marital status on men's exposure to mass media FP messages was completely attenuated in multivariate analysis. Similarly, we found that the difference between Christian and Muslim men completely attenuated in multivariate analysis; however, traditionalist men (OR = 0.50, $p < 0.01$) were still less likely to have been exposed to mass media than their Christian counterparts. The impact of structural factors on men's exposure to mass media FP messages remained largely robust in multivariate analysis. Men in South South (OR = 0.43, $p < 0.001$), South

**Table 1  Sample characteristics.**

|  | Percentage |
| --- | --- |
| Exposure to mass media FP messages | |
| No | 55 |
| Yes | 45 |
| Age | |
| 55–59 | 5 |
| 50–54 | 6 |
| 45–49 | 9 |
| 40–44 | 12 |
| 35–39 | 14 |
| 30–34 | 13 |
| 25–29 | 12 |
| 20–24 | 11 |
| 15–19 | 18 |
| Marital status | |
| Never married | 46 |
| Currently married | 53 |
| Formerly married | 1 |
| Religion | |
| Christian | 37 |
| Islam | 62 |
| Traditionalist | 1 |
| Region of residence | |
| South East | 12 |
| South South | 12 |
| South West | 19 |
| North Central | 14 |
| North East | 16 |
| North West | 27 |
| Place of residence | |
| Urban | 46 |
| Rural | 54 |
| Education | |
| Higher education | 17 |
| Secondary education | 46 |
| Primary education | 14 |
| No education | 23 |
| Household income | |
| Highest | 23 |
| Higher | 21 |
| Middle | 20 |
| Lower | 18 |

**Table 1** (*continued*)

|  | Percentage |
|---|---|
| Lowest | 18 |
| Employment status |  |
| Employed | 87 |
| Unemployed | 13 |
| Total | 13,294 |

West (OR = 0.81, $p < 0.05$), North Central (OR = 0.28, $p < 0.001$), North East (OR = 0.69, $p < 0.001$), and North West (OR = 0.81, $p < 0.05$) were all less likely to have been exposed to mass media FP messages compared to those in South East. Similarly, rural men (OR = 0.80, $p < 0.001$) were less likely to have been exposed to mass media FP messages than their urban counterparts. Furthermore, men with secondary (OR = 0.61, $p < 0.001$), primary (OR = 0.54, $p < 0.001$), and no education (OR = 0.34, $p < 0.001$) were less likely to have been exposed to mass media FP messages than their counterparts with higher education. Similarly, men whose household income belongs to the higher (OR = 0.84, $p < 0.01$), middle (OR = 0.57, $p < 0.001$), lower (OR = 0.47, $p < 0.001$), or lowest (OR = 0.25, $p < 0.001$) categories were less likely to have been exposed to mass media FP messages than their highest counterparts. Unemployed men (OR = 0.59, $p < 0.001$) were also less likely to have been exposed to mass media FP messages than employed men.

# DISCUSSION

Research shows the importance of men's involvement in FP decision making within households. Yet, we know very little about men's exposure to mass media FP messages in Nigeria although these messages are considered useful for increasing the uptake of FP among couples. To address this void in the literature, we adopted the structural influence model of health communication and used the Nigeria Demographic and Health Survey to explore the factors associated with men's exposure to FP massages through mass media. Results from the descriptive analysis show that as much as 55% of men had not been exposed to mass media FP messages through any of the four media platforms. In comparison to other countries in SSA, however, the proportion of unexposed men might be slightly lower in Nigeria. *Abita & Girma (2022)*, for example, found more than 65% of men in Ethiopia had not been exposed to mass media FP messages. Compared to women in Nigeria, men appear to have higher exposure with some evidence suggesting as much as 67% of women had not been exposed to mass media FP messages (*Ajaero et al., 2016*). The relatively higher awareness among men in Nigeria may likely be due to the intensified policies on FP in the last decade, such as the *Nigeria Family Planning Blueprint* that have worked to expose men to FP messages using mass media channels.

In the multivariate analysis, we found that men exposed to mass media FP messages were more likely to use FP than their unexposed counterparts. This observation is consistent with results of previous studies in the African context (*Abita & Girma, 2022*; *Do et al., 2020*; *Mutumba, 2022*). Given the pervasiveness of patriarchal norms in sub-Saharan Africa including Nigeria where men as household heads play a crucial role in household

**Table 2** Bivariate and multivariate analysis of mass media exposure to FP messages.

| | Bivariate analysis | | | | Multivariate analysis | | | |
|---|---|---|---|---|---|---|---|---|
| | OR | p-value | 95% CI | | OR | p-value | 95% CI | |
| Age | | | | | | | | |
| 55–59 | 1.00 | | | | 1.00 | | | |
| 50–54 | 1.19 | 0.210 | 0.91 | 1.54 | 1.09 | 0.550 | 0.82 | 1.45 |
| 45–49 | 1.02 | 0.869 | 0.80 | 1.30 | 0.86 | 0.268 | 0.65 | 1.12 |
| 40–44 | 1.07 | 0.567 | 0.85 | 1.35 | 0.85 | 0.200 | 0.65 | 1.09 |
| 35–39 | 1.00 | 0.989 | 0.79 | 1.26 | 0.79 | 0.066 | 0.61 | 1.02 |
| 30–34 | 0.92 | 0.455 | 0.73 | 1.15 | 0.72 | 0.012 | 0.56 | 0.93 |
| 25–29 | 0.74 | 0.011 | 0.58 | 0.93 | 0.64 | 0.001 | 0.49 | 0.84 |
| 20–24 | 0.52 | 0.000 | 0.41 | 0.65 | 0.47 | 0.000 | 0.35 | 0.63 |
| 15–19 | 0.28 | 0.000 | 0.22 | 0.35 | 0.29 | 0.000 | 0.21 | 0.39 |
| Martial status | | | | | | | | |
| Never married | 1.00 | | | | 1.00 | | | |
| Currently married | 1.91 | 0.000 | 1.75 | 2.09 | 0.97 | 0.694 | 0.92 | 1.20 |
| Formerly married | 1.38 | 0.061 | 0.98 | 1.94 | 0.87 | 0.504 | 0.72 | 1.42 |
| Religion | | | | | | | | |
| Christian | 1.00 | | | | 1.00 | | | |
| Muslim | 0.62 | 0.000 | 0.57 | 0.68 | 0.99 | 0.996 | 0.88 | 1.14 |
| Traditionalist | 0.39 | 0.000 | 0.24 | 0.63 | 0.50 | 0.008 | 0.30 | 0.84 |
| Region of residence | | | | | | | | |
| South East | 1.00 | | | | 1.00 | | | |
| South South | 0.45 | 0.000 | 0.38 | 0.52 | 0.43 | 0.000 | 0.36 | 0.52 |
| South West | 0.98 | 0.826 | 0.84 | 1.15 | 0.81 | 0.000 | 0.68 | 0.96 |
| North Central | 0.22 | 0.000 | 0.19 | 0.26 | 0.28 | 0.040 | 0.23 | 0.33 |
| North East | 0.31 | 0.000 | 0.27 | 0.37 | 0.69 | 0.000 | 0.56 | 0.84 |
| North West | 0.39 | 0.000 | 0.34 | 0.45 | 0.81 | 0.015 | 0.66 | 0.99 |
| Place of residence | | | | | | | | |
| Urban | 1.00 | | | | 1.00 | | | |
| Rural | 0.38 | 0.000 | 0.35 | 0.42 | 0.80 | 0.000 | 0.72 | 0.89 |
| Education | | | | | | | | |
| Higher | 1.00 | | | | 1.00 | | | |
| Secondary | 0.42 | 0.000 | 0.37 | 0.47 | 0.61 | 0.000 | 0.53 | 0.70 |
| Primary | 0.39 | 0.000 | 0.33 | 0.46 | 0.54 | 0.000 | 0.45 | 0.65 |
| No education | 0.17 | 0.000 | 0.14 | 0.19 | 0.34 | 0.000 | 0.29 | 0.41 |
| Household wealth | | | | | | | | |
| Highest | 1.00 | | | | 1.00 | | | |
| Higher | 0.67 | 0.000 | 0.58 | 0.77 | 0.84 | 0.020 | 0.73 | 0.97 |
| Middle | 0.40 | 0.000 | 0.35 | 0.45 | 0.57 | 0.000 | 0.49 | 0.67 |
| Lower | 0.28 | 0.000 | 0.25 | 0.32 | 0.47 | 0.000 | 0.39 | 0.55 |
| Lowest | 0.13 | 0.000 | 0.12 | 0.16 | 0.25 | 0.000 | 0.20 | 0.31 |

**Table 2** (*continued*)

|  | Bivariate analysis | | | Multivariate analysis | | |
|---|---|---|---|---|---|---|
|  | OR | *p*-value | 95% CI | | OR | *p*-value | 95% CI | |
| Employment | | | | | | | | |
| Employed | 1.00 | | | | 1.00 | | | |
| Unemployed | 0.46 | 0.000 | 0.40 | 0.52 | 0.59 | 0.000 | 0.50 | 0.70 |

decision making such as reproductive health decisions, engaging men in FP programs as well as women might be more beneficial than targeting only women. These messages could be tailored to encourage men to engage in discussions with their partners about family planning and to promote the use of contraception as a shared responsibility. Engaging men in FP policy programs through mass media in Nigeria could help reduce the country's higher maternal and neonatal burden.

In line with the structural influence model of health communication, we found a range of structural factors were associated with men's exposure to mass media FP messages. For example, we found that men with less than higher education were less likely to be exposed to mass media FP messages than their counterparts with higher education. This finding is consistent with previous research (*Babalola et al., 2015*; *Dougherty et al., 2018*; *Mutumba, 2022*), suggesting that education is a significant predictor of men's exposure to mass media FP messages. Men with higher levels of education may be more informed about the benefits of FP, potentially leading them to seek relevant mass media information on how to access and utilize FP. In addition, research points to the importance of English language proficiency on understanding mass media health communication in some SSA countries including Nigeria (*Okigbo et al., 2015*). In this regard, it is possible that educated men may personally feel targeted by mass media health campaigns presented in the official English language relative to their counterparts with lower levels of education who face challenges reading and digesting these messages.

We also found that poorer and unemployed men were less likely to be exposed to FP messages on mass media, compared to their richer and employed counterparts. These results are corroborated by earlier studies (*Aboagye et al., 2021*; *Ahmed et al., 2019*; *Konkor et al., 2019*), which argue that financial and material resources play a critical role in seeking health care information through mass media including FP information. In Nigeria, access to mass media may attract some financial burdens that may be overwhelming for socioeconomically disadvantaged groups such as low income households and the unemployed (*Ahmed et al., 2019*). Given the evidence by the World Bank that 4 in 10 people in Nigeria live below the poverty line, these groups may be preoccupied with meeting basic household needs such as shelter and food (*World Bank, 2022*), making it difficult to prioritize meeting the direct and indirect costs associated with accessing health information including mass media FP messages. Due to these contextual dynamics, economically disadvantaged groups may be facing unique barriers in accessing mass media FP messages.

In addition to socioeconomic characteristics, other structural factors such as region or state of residence and rural–urban residency were associated with men's exposure to mass media FP messages. Specifically, we found that rural men were less likely to be exposed to

FP messages than those in urban areas. This observation can be explained by the urban advantage in the location of mass media infrastructure. For example, the Nigerian Urban Health Reproductive Initiative—one of the major FP mass media initiatives in Nigeria—is concentrated in urban areas such as Ibadan and Kaduna (*Babalola et al., 2015*), possibly leaving behind rural residents who continue to have limited available options for mass media FP messages. These locational dynamics may be extended to the observed differences linked to region of residence. For example, given the rurality of North Central Region of Nigeria, residents within this region are less likely to be exposed to mass media FP messages compared to their counterparts in more urbanized South East Region (*Abubakar et al., 2022*).

Moreover, as suggested by the structural influence model of health communication, some mediating and moderating factors were associated with men's exposure to mass media FP messages. For example, age was a significant predictor of men's exposure to FP messages through mass media, indicating that younger age cohorts of men were less likely to be exposed than the oldest cohort. Elaborating this finding, earlier studies (*Aboagye et al., 2021*; *Kpienbaareh et al., 2022*) have pointed to older age as a need factor for accessing and utilizing health care services including mass media health communication. In this regard, older cohorts of men may particularly be prone to FP message exposure as they actively seek information through mass media on improving their health and that of dependants. In addition, we also found that traditionalist men were less likely to be exposed to mass media FP messages in comparison to their Christian counterparts. This result is consistent with previous research that alludes to the importance of large family size to members of traditional religion in Nigeria (*Babalola et al., 2015*; *Barro & Bado, 2021*). Specifically, the desire for more children as a sacred command and religious obligation may work to reduce their conscious and deliberate effort to seek FP messages through mass media (*Akintunde, Lawal & Simeon, 2013*).

There may be some limitations associated with this study worth highlighting. First, the NDHS is cross-sectional in nature, which limits our findings to statistical associations. The findings should therefore be interpreted with caution as they do not imply causal relationships. In addition, it is important to note that we relied on self-reported measures to construct men's exposure to mass media FP messages, which may be subjected to recall bias. Also, we did not include the Internet as one of the media channels in our study due to the lack of relevant data. We recommend the NDHS and other future studies on mass media exposure to include the internet as one of key sources of mass media messaging. Furthermore, due to the sensitive nature of reproductive and sexual behaviours in SSA, some respondents may under-report their exposure to FP messages through mass media. To address these limitations, there is a need for qualitative research to further explore in-depth nuances and lived experiences of men's exposure to mass media FP messages in Nigeria. Despite these limitations, this study provides useful insights for understanding men's exposure to mass media FP messages in the patriarchal context of Nigeria and elsewhere in SSA.

## CONCLUSION

Based on our findings, there are several policy implications and recommendations. For example, we observed that men with lower levels of education and household wealth as well as unemployed men were less likely to be exposed to mass media FP messages. As a policy response, addressing socioeconomic inequalities may be a critical approach to increasing couples' uptake of FP as a result of men's involvement in reproductive decision making through their exposure to mass media FP messages. However, in the short term, community-level interventions such as peer-to-peer education may be a useful and effective approach to reaching socioeconomically disadvantaged men and their partners with FP messages. We also found that some geographical characteristics such as rurality impacted men's exposure to mass media FP messages. To address these disparities, we recommend incentivising mass media outlets to extend their services and coverages to rural areas as well as the North Central region of Nigeria. As an immediate intervention, it may be also important to strengthen and resource existing health care infrastructure in rural areas to provide FP messages as part of their mandate. Moreover, some demographic characteristics such as age and religious affiliation were found to be significant predictors of men's exposure to mass media FP messages. It is critical for stakeholders to craft FP messages to be more appealing and culturally sensitive to younger cohorts of men as well as those who belong to traditional religion. For young men in particular, there is an urgent need to deliver FP messages through more modern mass media outlets including social media and other online-based platforms.

### Funding
The authors received no funding for this work.

### Competing Interests
The authors declare there are no competing interests.

### Author Contributions
- Daniel Amoak conceived and designed the experiments, performed the experiments, analyzed the data, prepared figures and/or tables, authored or reviewed drafts of the article, and approved the final draft.
- Irenius Konkor analyzed the data, authored or reviewed drafts of the article, and approved the final draft.
- Kamaldeen Mohammed performed the experiments, analyzed the data, prepared figures and/or tables, authored or reviewed drafts of the article, and approved the final draft.
- Sulemana Ansumah Saaka analyzed the data, authored or reviewed drafts of the article, and approved the final draft.
- Roger Antabe conceived and designed the experiments, performed the experiments, analyzed the data, prepared figures and/or tables, authored or reviewed drafts of the article, and approved the final draft.

## Data Availability

The Nigeria DHS, 2018 report is available at The DHS Program:
https://www.dhsprogram.com/publications/publication-fr359-dhs-final-reports.cfm.

## Supplemental Information

Supplemental information for this article can be found online at http://dx.doi.org/10.7717/peerj.15391#supplemental-information.

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
