# Peer review of "Exposure to mass media family planning messages among men in Nigeria: analysis of the Demographic and Health Survey data"

_PeerJ, doi:10.7717/peerj.15391_

## Round 0.1 · original submission · Minor Revisions

I suggest that you look at each of the points that the reviewers are making. You may choose to incorporate a few suggestions including in limitation the inability to depict data about FP messages from the internet or as well respond to certain other queries and comments.

I look forward to your revised submission.

Reviewer 1 ·

Basic reporting

See comment below.

Experimental design

See comment below.

Validity of the findings

See comment below.

Additional comments

Thank you very much for sharing this manuscript with me. I believe this topic is very important. I have several minor comments.

1. I see that the authors rely on FP messages from four different media channels including radio, television, text messages, and print media. How about the Internet? Did you not include this due to a lack of relevant data? If so, I would suggest mentioning this in the limitation section.

2. This study finds 45% of men in Nigeria are exposed to mass media FP messages. What does this estimate mean? Is this higher or lower compared to men in other SSA countries? Is this estimate higher for women than men? It may be useful to discuss and contextualize this estimate in the discussion section.

3. There are several minor style and grammar issues throughout the manuscript. Please carefully check them again.

·

Basic reporting

In this manuscript, the authors report on statistical correlations between socioeconomic and geographical measures and exposure of family planning initiatives in men in Nigeria. The article raises and important question, and does reasonable analysis to answer that question. I recommend accepting this article with a few minor revisions:

1. The most important edit required is in line 206 where the authors report that "that 55% of men have been exposed to FP messages through at least one of the four mass media outlets". However, in Table 1, the first couple rows state that 55% responded "No" to the question about Exposure to mass media FP. It's unclear whether the text is wrong or the table, but at the moment they are inconsistent.

2. Please give a few examples of the FP methods referred to on line 78

3. Line 115: Change "educated people may better position to digest health information" to "educated people may be better positioned to digest health information" to fix the grammar error

4. Line 159: Change "sampling with probability to size" to "sampling with probability proportional to size"

5. Line 217 is the first place where OR is used as an acronym, but it is never expanded. I guess it implies Odds ratio, but please clarify that in the text, specifically in the Methods when Odds ratio is being described first

6. Line 251: Change "men whose household income belongs the higher, middle, lower or lowest categories" to "men whose household income belongs to the higher, middle, lower or lowest categories"

Experimental design

The analyses done in this manuscript are simple but clear. The methods are reasonable. A soft suggestion would be to make a graphical illustration of some of their important claims.

Validity of the findings

The findings listed in the Results and described in detail in the Discussion are valid. They also lead to conclusions that are highly relevant and important to targeting relevant populations for improved family planning media exposure

Reviewer 3 ·

Basic reporting

The manuscript is well written and structured. I commend the author's efforts.
An abstract that is arranged in a structured manner briefly describes the article contains clearly. The Introduction section is well written, placing the present research context. Also, the author(s) mentions notable research objectives. The author(s) also tried to present comprehensive literature in the literature review section. However, this kind of research has been widely carried out, especially in African countries. Then, what is unique about this research? The authors need to emphasize the uniqueness of this research.

Experimental design

In the measurement section, the author measures "Media Exposure" by asking whether they are exposed or not, which is possible the audience may forget whether he was exposed or not. Is it not problematic and accurate enough to measure?.

Validity of the findings

The results section looks fine to me. The discussion section is well-written. The authors reported the implications. The author(s) has explained the limitations of this research well.

Additional comments

No comment

---

## Round 0.2 · accepted · Accept

The reviewers have submitted their report and all have been satisfied with the revision made to address their comments. Your manuscript is accepted.

Reviewer 1 ·

Basic reporting

See my comments below.

Experimental design

See my comments below.

Validity of the findings

See my comments below.

Additional comments

All my comments have been addressed.

·

Basic reporting

The authors have addressed my comments from my previous review. Thank you for incorporating feedback. I also think that the authors have sufficiently addressed other reviewers' concerns as well

Experimental design

n/a

Validity of the findings

n/a

Reviewer 3 ·

Basic reporting

No Comment

Experimental design

No Comment

Validity of the findings

No Comment

Additional comments

The author(s) has made strong revisions.